# MECHANISTIC NEURAL NETWORKS

## ABSTRACT

We present Mechanistic Neural Networks, a new neural module that represent the evolution of its input data in the form of differential explicit equations. Similar to regular neural networks, Mechanistic Neural Networks $F(x)$ receive as input system observations $x$, *e.g.* $n$-body trajectories or fluid dynamics recordings. However, unlike regular neural network modules that return vector-valued outputs, mechanistic neural networks output (the parameters of) a *mechanism* $\mathcal{U}_x = F(x)$ in the form of an explicit symbolic ordinary differential equation $\mathcal{U}_x$ (and not the numerical solution of the differential equation), that can be solved in the forward pass to solve arbitrary tasks, supervised and unsupervised. Providing explicit equations as part of multi-layer architectures, they differ from Neural ODEs, UDEs and symbolic regression methods like SINDy. To learn explicit differential equations as representations, Mechanistic Neural Networks employ a new parallel and differentiable ODE solver design that (i) is able to solve large batches of independent ODEs in parallel on GPU and (ii) do so for hundreds of steps at once (iii) with *learnable* step sizes. The new solver overcomes the limitations of traditional ODE solvers that proceed sequentially and do not scale for large numbers of independent ODEs. Mechanistic Neural Networks can be employed in diverse settings including governing equation discovery, prediction for dynamical systems, PDE solving and yield competitive or state-of-the-art results.

## 1 INTRODUCTION

Understanding and modeling the mechanisms underlying the evolution of data is a fundamental scientific challenge and is still largely performed by hand by domain experts, who leverage their understanding of natural phenomena to obtain equations. This process can be time-consuming, error-prone, and limited by prior knowledge. Towards automating this, causal models can answer certain interventional and counterfactual questions, but they still do not allow learning explicit mechanistic models providing physical insights, see Table 1 in Schölkopf et al. (2021). In this paper, we introduce *Mechanistic Neural Networks*, an approach designed to learn explicit governing mechanisms from data.

Mechanistic Neural Networks directly generate equations for approximating the mechanisms governing the evolution of data rather than the scalar-valued or vector-valued outputs of vanilla neural networks. Our approach is close in spirit to the scientific standard of modeling system observations by specifying governing equations. Given an observation datum $x$ or a sequence of observations $x = [x_1, ..., x_t]$ (e.g. $n$-body trajectories), a Mechanistic Neural Network outputs an *explicit* mechanism $\mathcal{U}_x = F(x)$ that characterizes the patterns in the datum $x$. These output equations are in the form of ordinary differential equations with arbitrary coefficients controlled by non-linear neural networks. Subsequently, an input evolves according to its generating mechanism to produce a set of intermediate and final states for downstream processing.

Our approach is in stark contrast with Neural Ordinary Differential Equations (Neural ODEs (Chen et al., 2018)), which learn a global (implicit) model for a given system. While with Neural ODEs there is an ODE by the end of learning, that is in the form of a highly-parameterized neural network and thus not really interpretable. By contrast, the learned differential equations in Mechanistic Neural Networks can be *explicitly and compactly written down*, providing *interpretable and analyzable mechanisms*. The explicit form of the equations enhances their utility in various scientific domains, where insights and explanations are crucial. Our approach is also very different from symbolic regression methods like SINDy (Brunton et al., 2016), which rely only on linear regression, cannot scale easily to more complex and non-linear dynamical settings, and cannot be integrated as layers in

neural network architectures. By contrast, Mechanistic Neural Networks can leverage the power of non-linear neural networks while still deriving explicit mechanisms, thus being able to capture the richness and expressiveness of complex dynamical systems.

To efficiently learn and solve the generated mechanisms, Mechanistic Neural Networks propose a new scalable, parallel and differentiable ODE solver. Mechanistic networks require fast parallel solution of large batches of independent ODEs. Traditional solvers such as Runge-Kutta methods are too inefficient for this scale due to their sequential processing of steps and their inability to handle independent ODEs in parallel. Our proposed solver has (i) *step parallelism*, i.e., it can solve for hundreds for steps in parallel and (ii) *batch parallelism*, where independent ODEs can be solved in parallel on GPU.

To assess the effectiveness of Mechanistic Neural Networks, we experiment in an extensive array of problem settings simultaneously and compare with state-of-the-art that can only focus on single problem settings, including governing equations discovery (*e.g.,* with SINDy), solving PDEs (*e.g.,* with FNO), and dynamical modeling (*e.g.,* with Neural ODE). Results demonstrate we can produce mechanisms that accurately model complex dynamical processes, showcasing the power, versatility, and generality of Mechanistic Neural Networks in capturing and understanding the evolution of data.

## 2 MECHANISTIC NEURAL NETWORKS

We aim to learn neural networks that *represent explicit differential equations* prescribing evolution of data in internal representation, rather than returning vectors like regular neural networks. We call such models *Mechanistic Neural Networks* (MNN). The representations in MNNs rely on ordinary differential equations of arbitrary order.

Formally, a MNN represents a function $F$ that takes an input $x$ and produces a differential equation $\mathcal{U}_x$. The high-level symbolic form of an MNN can be given as

$$\mathcal{U}_x = F(x) \tag{1}$$

where $\mathcal{U}_x$ is a $d$th order parameterized ODE over $u = u(t)$ with time-varying coefficients. The ODE, in general, may depend on the input datum $x$ and is written

$$\mathcal{U}_x : \overbrace{\sum_{i=0}^{d} c_i(t; x) u^{(i)}}^{\text{Linear Terms}} + \overbrace{\sum_{k=0}^{r} g_k(t, u, u', \ldots; \phi_k(t; x))}^{\text{Non-Linear Terms}} = b(t; x). \tag{2}$$

where we specify linear and non-linear terms separately. Linear terms with derivatives $u^{(i)}$ have coefficients $c_i(t; x)$ and $b(t; x)$ that are arbitrary non-linear functions of time. $g_k$ are an arbitrary number of non-linear terms in $u^{(k)}$ that may have time-dependent parameters $\phi_k(t; x)$ of their own. Note that $u$ may be multidimensional and equation 2 would then be a system of ODEs, however we assume a single dimension for exposition.

**Neural Network instantiation.** While having described the general form of MNNs, we still need to instantiate specifically the neural network parameterization of equation 1, which is $w_x = f_\theta(x)$. $f$ is a neural network with parameters $\theta$ and $w_x = [c_{i,t}, \phi_{k,t}, b_t, s_t]$ represents a concrete parameterization of the ODE $\mathcal{U}_x$ on an $n$-step time grid with step sizes $s_t$, where $t$ is now a *time index* with slight abuse of notation. With MNNs the grid discretization can also be learned by learning the step size $s_t$ for each time step. We convert the ODE of equation 1 to a discrete ODE

$$\sum_{i=0}^{d} c_{i,t} u_t^{(i)} + \sum_{k=0}^{r} g_k(t, u_t, u'_t, \ldots; \phi_{k,t}) = b_t \quad \text{with steps } s_t, \tag{3}$$

where $c_{i,t}$ represent the discretized coefficient $c_i$ for time index $t = 1, \ldots, n$, and $u_t, u'_t, u''_t, \ldots, u_t^{(d)}$ represent the discretized function and derivative values at $t$. For simplicity we do not specify initial conditions which are optional in our setting and can also be learned. We note that, in general, *one ODE is generated for a single data example $x$* and a different example $x'$ would lead to a different ODE.

Having described how MNN parameterize the ODE hidden representation, we now describe the forward propagation, backpropagation, and training procedures. Making predictions efficiently requires solving the ODE in a differentiable way that is also (1) parallelizable on GPU for multiple samples, (2) can tackle hundreds of steps at once, and (3) allow learnable step sizes.

## 2.1 Forward propagation and prediction

**Generating the ODE $\mathcal{U}_x$.** Forward propagation in MNNs to generate the ODE representation is straightforward, as $f_\theta(x)$ produces the parameters for the ODE $w_x = f_\theta(x)$. This is the general form of an MNN which gives an ODE per input. However, we can also generate *global* ODEs by sharing ODE coefficients between different examples.

Overall, to specify the $n$-step discretization of a one-dimensional ODE from equation 2 with $r$ non-linear terms requires specifying of $d + r + n + 1$ parameters *per time step*: $d + 1$ parameters specify the values of $c_{i,t}$ for orders $i = 0, ..., d$ (including the 0-th order for the function evaluation) on the left-hand side, one parameter for $b_i$ on the right-hand side, $r$ parameters for nonlinear terms and $n - 1$ step sizes.

**Making predictions.** Mechanistic Neural Networks produce as output for a given input $x$ a parameterized ODE $\mathcal{U}_x$ (or a set of ODEs) as described by equation 2 (or, more precisely, the discrete variant in equation 3). Next we solve the ODE using a specialized solver, obtaining solution $u$ (and all derivatives up to order $d$) discretized over the grid, that is

$$[u_t^d, \dots, u_t] = \texttt{ODESolve}(w_x, u_{\text{init}}), \qquad (4)$$

where $u_{init}$ is an initial condition which may be set from data or it may be learned.

**The final output** is obtained by feeding the solution into an optional decoder model with $y = \text{NN}(u_t^d, \dots, u_t)$. During training, we optimize parameters $\theta$ for the neural network $f_\theta$, producing the ODE coefficients $w_x$, and the decoder jointly by minimizing a loss $\ell(y, y^*)$, where $y$ is a prediction and $y^*$ is a target. The training signal is generic and can cover both supervised and unsupervised learning. In the experiments, we use the MSE loss for regression and cross-entropy loss for classification. $y^*$ can be a future target, the input $x$ itself (as in governing equation discovery), or a classification target.

Since during forward propagation there is an ODE solver step, we need to compute gradients through the solver. MNNs can produce a large number of ODEs. In the next section, we describe our mathematical programming proposal for efficiently solving ODEs in parallel.

## 2.2 Efficient Backpropagation and training

**Main challenge.** The forward and backward passes require a significant amount of computation for solving ODEs *en masse* and for computing gradients. General purpose solvers for solving ODEs such as Runge-Kutta methods are *sequential*, proceeding one step at a time. Furthermore, they are very inefficient for solving batches of *independent ODEs*, since each independent example requires an independent computation.

In the coming section, we present a novel efficient and parallel algorithm for solving batches of independent ODEs. We revisit an early proposal (Young, 1961) for representing linear ODEs as linear programs, thus reducing solving linear ODEs to solving linear programs. Next, we enhance the approach for solving *non-linear* ODEs and make it differentiable for our use in neural network representations. This is done with a differentiable equality-constrained quadratic programming relaxation for solving the ODEs in the forward pass and differentiable optimization techniques for obtaining gradients in the backward pass. This approach for solving ODEs allows efficient solution of batches of ODEs in parallel with GPU acceleration. Furthermore, the solution is obtained for a large number of steps in parallel, significantly improving over the sequential approach of traditional solvers.

## 3 Differentiable Optimization for Solving and Learning ODEs

In this section we propose a new differentiable ODE solver using equality-constrained quadratic programming enhancing an early linear programming solver for *linear* ODEs (Young, 1961). The proposed solver allows fast parallel solution of batches of ODEs with parallel steps, has a *learnable* discretization, and allows differentiation of ODE solutions relative to ODE parameters required for mechanistic networks. Unlike the solver from Young (1961), when combined with neural networks the QP solver can solve non-linear ODEs.

### 3.1 Solving ODEs with Quadratic Programming

A general equality-constrained quadratic program is specified by a quadratic objective to be minimized and a set of linear equality constraints.

$$
\begin{aligned}
\text{minimize} \quad & \tfrac{1}{2} z^t G z + \delta^t z \\
\text{subject to} \quad & A z = \beta,
\end{aligned}
\tag{5}
$$

where $A \in \mathbb{R}^{l \times n}$ is the constraint matrix with $l$ constraints, $\beta \in \mathbb{R}^l$, $G \in \mathbb{R}^{n \times n}$ and $\delta \in \mathbb{R}^n$. We reduce an ODE solution to solving an equality-constrained quadratic program.

Equality-only quadratic programs can be solved significantly more efficiently than general quadratic programs with inequality constraints. We describe the quadratic programming formulation of our ODE solver by specifying the variables, constraints and objective.

**Variables.** For the linear terms (equation 2), we introduce one variable per function value $u_t$ and per $k = 1, ..., d$-order derivative $u'_t, u''_t, \ldots, u_t^{(d)}$ and for all time indices $t = 1, \ldots, n$. For each non-linear term $k$, we add variables $\nu_{k,t}, \nu'_{k,t}, \ldots, \nu_{k,t}^{(d)}$ for the function values and derivatives of the non-linear term for each time step. We add one variable for the error $\epsilon$.

**Constraints.** We have three types of constraints: *equation constraints*, *initial-value constraints*, and *smoothness constraints*. *Equation constraints* specify that at each time step $t$, the left-hand size of the discretized ODE is equal to the right-hand side. For instance, if we are interested in output functions described by a second order ODE, we would have one equality constraint

$$
c_{2,t} u''_t + c_{1,t} u'_t + c_{0,t} u_t = b_t, \forall t \in \{1, \ldots, n\}.
\tag{6}
$$

*Initial-value constraints* specify constraints on the function or its derivatives for the initial conditions at $t = 0$. For a second order ODE type of output, for instance, we may specify that the initial function and first derivative are equal to 0,

$$
u_0 = 0, \quad u'_0 = 0
\tag{7}
$$

*Smoothness constraints* ensure that the function and its derivative values at adjacent steps vary smoothly. These constraints associate the function and derivative variables at each time step to neighbouring function and derivatives variables by Taylor approximations. Smoothness constraints state that the error in the Taylor approximations should be equal to the allowed error $\epsilon$.

For the forward and backward Taylor approximation $u(t \pm s_t)$ and $u'(t \pm s_t)$ for step $s_t$, we add constraints on the function and derivative variables $u_t, u'_t, \ldots$ and $\nu, \nu', \ldots$ as follows,

$$
\left.
\begin{aligned}
u_t + s_t u'_t + \tfrac{1}{2} s_t^2 u''_t - u_{t+1} &= \epsilon \\
s_t u'_t + \tfrac{1}{2} s_t^2 u''_t - s_t u'_{t+1} &= \epsilon
\end{aligned}
\right\} \quad \text{Forward Taylor approximation}
\tag{8}
$$

$$
\left.
\begin{aligned}
u_t - s_t u'_t + \tfrac{1}{2} s_t^2 u''_t - u_{t-1} &= \epsilon \\
-s_t u'_t + \tfrac{1}{2} s_t^2 u''_t + s_t u'_{t-1} &= \epsilon
\end{aligned}
\right\} \quad \text{Backward Taylor approximation}
\tag{9}
$$

We add similar constraints for derivatives of all orders. We collect the constraints from equations 6, 7, 8, 9 into a constraint matrix $A$ and the right hand sides into $\beta$ in 5. Note that we use the same scalar error variable $\epsilon$ for all time steps and orders in equations 8 and 9.

**Objective.** The objective of the quadratic program is to minimize $\epsilon^2$ which minimizes the error in the smoothness constraints. To make the quadratic program continuously differentiable relative to the objective and constraint parameters we relax the quadratic program by including a diagonal convex quadratic term in the objective, $G = \gamma I$ (Wilder et al., 2019) where $\gamma$ is a relaxation parameter chosen from $[0.1, 1]$.

**Nonlinear ODEs.** Solving the quadratic program alone is enough to solve linear ODEs. To solve non-linear ODEs, for each non-linear term $g_k(u, u'', \ldots; x)$ we add a squared error loss term $(\nu_k - g_k)^2$ in the objective. The loss learns the parameters of $g_k$ to ensure that the variable $v_k$ is close to the required non-linear function of the solution.

## 3.2 Efficient Solution and Gradients for Equality-Constrained QP.

The main advantage of equality-constrained quadratic programs is that they can be directly solved in parallel on GPU (Pervez et al., 2023).

**Forward propagation and solving the quadratic program.** Having defined our quadratic program, we can solve it directly with well-known techniques (Wright and Nocedal, 1999) by solving the following KKT system for some $\lambda$ by simplifying the KKT matrix.

$$\begin{bmatrix} G & A^t \\ A & 0 \end{bmatrix} \begin{bmatrix} -z \\ \lambda \end{bmatrix} = \begin{bmatrix} \delta \\ -\beta \end{bmatrix} \xrightarrow{\text{Simplifying KKT}} \begin{bmatrix} \gamma I & A^t \\ A & 0 \end{bmatrix}^{-1} = \begin{bmatrix} C & E \\ E^t & F \end{bmatrix}, \tag{10}$$

where $C = \frac{1}{\gamma}(I - A^t(AA^t)^{-1}A)$, $E = A^t(AA^t)^{-1}$ and $F = -(AA^t)^{-1}$. Solving the simplified KKT system can be efficiently implemented in the forward and backward propagations with a Cholesky factorization followed by forward and backward solve to obtain the solution $z$.

**Backward propagation and gradients computation.** In the backward pass, we need to update the ODE parameters in the constraint matrix $A$ as given in equation 6.

The gradient relative to constraint matrix $A$ is obtained by computing $\nabla_A \ell(A) = \nabla_A z \nabla_z \ell(z)$, where $\ell(.)$ is our loss function and $z$ is a solution of the quadratic program.

We can compute the individual gradients using already established techniques for differentiable optimization, see Amos and Kolter (2017) for details. Briefly, computing the gradient requires solving the system equation 10 for with a right-hand side containing the incoming gradient $g$:

$$-\begin{bmatrix} G & A^t \\ A & 0 \end{bmatrix} \begin{bmatrix} d_z \\ d_\lambda \end{bmatrix} = \begin{bmatrix} g \\ 0 \end{bmatrix}. \tag{11}$$

Since this uses the same KKT matrix as the forward solve we can reuse the Cholesky factorization of $AA^t$ for the gradient making the gradient computation very efficient. The gradient relative to $A$ can then be computed by first solving for $d_z, d_\lambda$ and then computing $d_\lambda z^t + \lambda d_z^t$ (Amos and Kolter, 2017).

### 3.3 Complexity

The computational and memory complexity of MNNs is determined by the size of the initial state $k$, the size of the time grid $n$, and the order $d$ of the ODEs to be generated. The last layer of $f$ to outputs $k \times n \times (d + 2)$ parameters. This means that the memory required to store the coefficients can be large depending on the size of the grid and the dimension. The main computational effort in solving the system equation 10 for a batch of ODEs, which we do by a Cholesky factorization and has cubic complexity in $nkd$. In practice, for our experiments, $n \approx 50$, $k \approx 100$, and a batch size of 30 have a reasonable memory and computational cost.

## 4 Related Work

**Neural Dynamical Systems.** In terms of data-driven modeling of dynamical systems with differential equations, MNNs are related to Neural ODEs (Chen et al., 2018). Instead of explicitly parameterizing the dynamics with differential equations, Neural ODEs represent dynamics as neural network which are then evolved forward in time by solving the resulting ODE by a solver embedded in the model. With neural ODEs the model can be seen as the forward evolution of a differential equation. With MNNs a set of ODEs are first generated and then solved in a single layer for a specified number of time steps. Furthermore, our solution of the ODE does not involve explicit forward propagation but requires solving a quadratic program. With our optimization of solving only equality-constrained quadratic programs the forward pass becomes very fast. Another important difference is that with Neural ODEs the learned equation is implicit and there is a single ODE for modeling the system. MNNs, on the other hand, explicitly generate the dynamical equation that governs the evolution of the input datum with potential for analysis and interpretation. Vanilla Neural ODEs can also be limited in ways that restricts their modeling capacity and variations such as augmentation (Dupont et al., 2019) and second-order ODEs (Norcliffe et al., 2020) overcome some of their limitations. MNNs generate a family of linear ODEs, one per initial state, with arbitrary order which makes them very flexible and enables non-linear modeling.

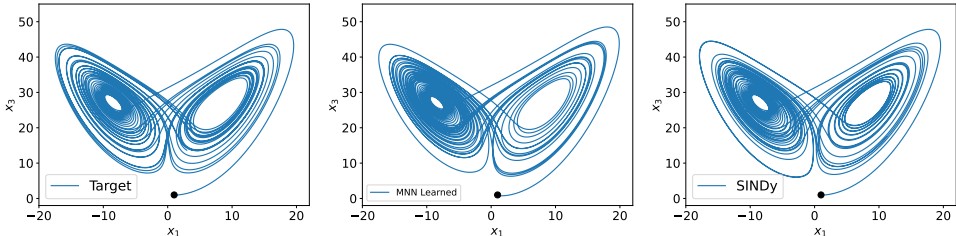

Figure 1: Solution of Learned ODEs for the chaotic Lorenz system

**ODE Solvers.** Traditional methods for solving ODEs involve numerical techniques such as finite difference approximations and Runge-Kutta algorithms. The linear programming approach to numerical solution of linear ordinary and partial differential equations was originally proposed around the 1960s (Young, 1961; Rabinowitz, 1968) shortly after the invention of linear programming (Dantzig, 1963). After a decade or two of activity, however, interest in the approach died down with little to no recent literature on the topic. MNNs require fast and GPU parallel solution to a large number of independent but simple linear ODEs for which general purpose solvers would be too slow. For this we revisit the linear program approach to solving ODEs, convert it to an equality constrained quadratic program for fast batch solving and resort to differentiable optimization methods (Amos and Kolter, 2017; Barratt, 2019; Wilder et al., 2019) for differentiating through the solver.

**Discovery.** MNNs are also related to prior work in discovery of physical mechanisms with observed data. SINDy (Brunton et al., 2016) discovers governing equations from time series data using a pre-defined set of basis functions combined with sparsity-promoting techniques. Symbolic regression is another class of methods for discovering symbolic mathematical expressions that fit observations usually implemented as evolutionary algorithms (Schmidt and Lipson, 2009). An advantage of MNNs is that they can handle larger amounts of data than shallow methods like SINDy. Other approaches to discover physical mechanisms are Physics-informed networks (PINNs) and universal differential equations (Rackauckas et al., 2020; Raissi, 2018), where we assume the general form of the equation of some phenomenon, we posit the PDE operator as a neural network and optimize a loss that enables a solution of the unknown parameters. In contrast MNNs parameterize a family of ODEs by deep networks and the solution is obtained by a specialized solver.

## 5 EXPERIMENTS

### 5.1 DISCOVERY OF GOVERNING EQUATIONS

We first experiment with discovering governing equations of linear and non-linear differential equations from data. We give the specific MNN architecture in Figure 11 in the appendix.

For discovering governing equations, we follow Brunton et al. (2016) and use differential equations of the form $u' = F(u)$, by specifying $F(u)$ in terms of a library of basis functions, such as polynomial functions up to some maximum degree denoted as $\Theta$ (see A.3 and Brunton et al. (2016) for details). The input to the basis functions is produced by a neural network from the input data. Learnable parameters $\xi$ specify the weight of each basis function by computing $\Theta\xi$. An arbitrary nonlinear function $g$ may be applied to $\Theta\xi$ to produce the right hand side of the differential equation as $u' = F(u) = g(\Theta\xi)$. Given the right hand side as $F(u)$ we use the QP solver to solve the ODE to produce the solution $u$. Given data $x$ we compute a two part loss function: The first part minimizes the MSE between $x$ and $u$ and the second part minimizes the MSE between the neural network output $f(x)$ and $x$. Using the neural network in this way allows for a higher capacity model and allows handling noisy inputs. The model is then trained using gradient descent. Following Brunton et al. (2016) we also threshold the learned parameters $\xi$ to produce a sparse ODE solution.

Note that unlike SINDy does not require using derivative computation using either finite differences or other related approaches since the solution is obtained by solving a differential equation. Furthermore, unlike SINDy the model is non-linear and can handle classes of differential equation where $F(u)$ depends non-linearly on $\Theta\xi$. In the following we show two examples of such cases where $F(u)$ is a rational function (a ratio of polynomials) and when $F(u)$ is a nonlinear function of $\Theta\xi$. Moreover, unlike SINDy, MNN can learn a single governing equation from multiple trajectories each with a different initial state making MNN more flexible. In many situations a single trajectory sample is not enough represent to the entire state space while multiple trajectories allow discovery of a more representative solution.

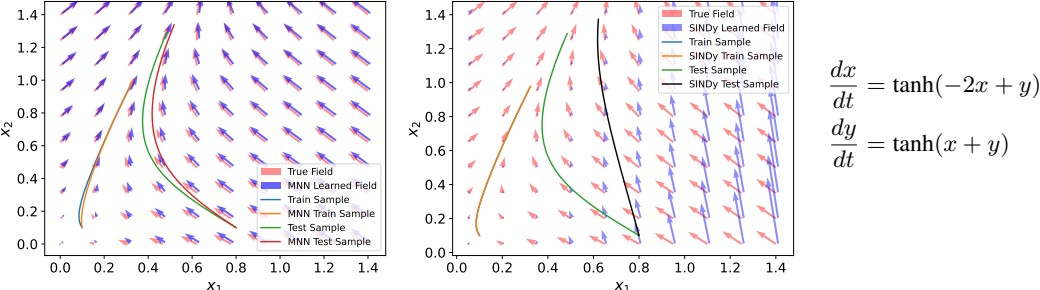

Figure 2: Learned ODE vector fields for MNN and SINDy with non-linear tanh function of basis combination and training and test trajectories. Ground truth equation is on the right.

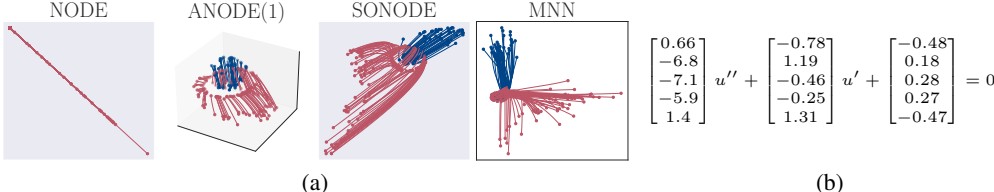

(a)                                                                                      (b)

Figure 3: (a) Visualizing the state evolution of the learned equations $\mathcal{U}_x$ data points in nested circles. The points from the two classes are perfectly separated despite the nested topology without requiring augmentations. (b) The learned equation $\mathcal{U}_x$ from MNN that corresponds to a single data point for 5 steps. This equation can thereafter be used for analysis, or even classification.

**Planar and Lorenz System.** We first examine the ability of MNN equation discovery for systems where the true ODE can be exactly represented as a linear combination of polynomial basis functions. We use a two variable planar system and the chaotic Lorenz system as examples. Both MNN and SINDy are able to recover the planar system. Simulation of the learned Lorenz ODE are shown in Figure 1 for MNN and SINDy.

Next we consider ODE systems where the derivative *cannot* be written as a linear combination of polynomial (or other) basis function.

**Nonlinear Function of Basis.** First, we consider systems where the derivative is given by a nonlinear function of a polynomial. For simplicity we assume that the nonlinear function is known. As an example we solve the system from figure 2 with the tanh nonlinear function.

Vector fields for the learned systems are shown in Figure 2 for MNN and SINDy. We see that although SINDy fits the training example, the directions diverge further away. With MNN we see that the learned vector field is consistent with the ground truth far from the training example even though we use only a single trajectory.

**Rational Function Derivatives.** Second, we consider the case where the derivative is given by a rational function, i.e., $F(u) = p(u)/q(u)$, where $p$ and $q$ are polynomials. Such functions cannot be represented by the linear combination of polynomials considered by SINDy, however such functions can be represented by MNNs by taking $p$ and $q$ to be two separate combinations of basis polynomials and dividing. An example is shown in Figure 12 in the appendix for the system where we see again MNNs learning much better equations compared to SINDy with a second-order polynomial basis tha overfits. Further, by including more trajectories in the training, results improve further, see Figure 12.

## 5.2 NESTED CIRCLES

We test MNN on the nested circles dataset (Dupont et al., 2019), where we must classify each particle as one of two classes. This task is not possible for unaugmented Neural ODEs since they are limited to differomorphisms (Dupont et al., 2019). We show the results in Figure 3, including comparisons with Neural ODE (Chen et al., 2018), Augmented Neural ODE and second-order Neural ODE (Norcliffe et al., 2020). MNNs can comfortably classify the dataset without augmentation and can also derive a governing equation.

## 5.3 AIRPLANE VIBRATIONS

MNNs can learn complex dynamical phenomena significantly faster than Neural ODE and second order Neural ODE. We reproduce an experiment with a real-world aircraft benchmark dataset (Noël

Figure 4: Modeling airplane vibrations. The estimated non-autonomous ODE in the further right.

and Schoukens, 2017; Norcliffe et al., 2020). In this dataset the effect of a shaker producing acceleration under a wing gives rise to acceleration $a_2$ on another point. The task is to model acceleration $a_2$ as a function of time using the first 1000 step as training only and to predicting the next 4000 steps. Results of the experiment are shown in Figure 4. We compare against Augmented Neural ODE and second order Neural ODE. MNNs are on-par with second-order ODE, converge significantly faster in the number of training steps, and achieve two times lower training error, showcasing the capacity for modeling complex phenomena and improving with modest architectural modifications. The predicted $a_2$ accelerations are very close to the true ones in the center-right plot.

## 5.4 PHYSICAL PARAMETER DISCOVERY WITH THE 2-BODY PROBLEM.

We perform a 2-body experiment with synthetic 2D data for physical parameter discovery. We generate two bodies with masses 10 and 20 and a gravitational force between the bodies, $F = G\frac{m_1 m_2}{r^2}\hat{r}$, with unit vector $\hat{r}$, where we set $G = 2$. We generate a trajectory with random initial positions and velocities by solving Newton's equation of motion with the gravitational force. In the 2-body case the ratio of the masses can be computed by taking the ratio of the accelerations since the forces are equal and opposite. First we verify that MNNs can indeed model the orbit. Figure 6 shows prediction for the orbit for MNN and Neural ODE. MNN has good prediction while Neural ODE is poor.

To learn the correct mass ratio, we note that the second law of motion only has a non-zero second derivative term and a force. To discover physical parameters we use a restricted MNN with ODEs that can represent Newton's laws. In the MNN model for this experiment, we use the same coefficient for the second derivative term for all time steps with the remaining coefficients fixed to 0.

We have a total of 4 coefficients representing the $x$ and $y$ coefficients for each body. For modeling the force, we use Newton's 3rd law, $F_{12} = -F_{21}$. Training the model trains a differential equation for the $x, y$ coordinates of each object of the form $c_{i,j}u''_{i,j} = F_t$ for $i = 1, 2$ objects, and $j$ is the $x$ or $y$ coordinate. The force $F$ is computed by neural networks satisfying the third law and superposition. After training, we find $c_{1,x} = 0.78, c_{1,y} = 0.86, c_{2,x} = 1.57, c_{2,y} = 1.83$ with the ratio of the coordinates $c_{1,x}/c_{2,x}$ and $c_{1,y}/c_{2,y}$ approximately 2.022 which is close to the true mass ratio of 2. Furthermore, we see in Figure 6 that the ground truth forces are close to the predicted forces after normalization .

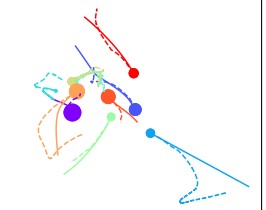

Figure 5: 10-body trajectory predictions (dashed lines are predictions). Trajectories are 4000 steps. The first half only are used for training.

## 5.5 N-BODY PREDICTION

For a more complex prediction task, we consider 10 interacting bodies with fixed masses and random initial positions and velocities. We consider the motion of the bodies under the influence of the gravitational force. We use a basic MNN with a second-order ODE to model the dynamics. We use half the trajectory for training and predict the latter half. The results are shown in Figure 5. MNN does model the global trajectory although finer details are lacking resulting in larger deviations for some particles. We show quantitative comparisons with Neural ODE in Table 2 which is unable to model this task.

## 5.6 PDE SOLVING

We next examine using an MNN module for building a PDE solving architecture. We use a 2d Darcy flow dataset with resolution 85x85 (Li et al., 2020c). We build the module as shown in

Figure 6: 2-body problem: Predicted orbits for MNN and Neural ODE, and normalized true and learned force vectors during 200 steps.

Figure 13 in the appendix. The module uses 2d convolutional layers where the feature map is downsampled and ODE parameters are computed and solved. The ODE is solved for 30 steps and the entire soluton trajectory is then upsampled and combined with the input features map. The network is built by stacking three such modules together plus an input MLP layer and an output layer.

### 5.7 QP Solver Evaluation

In this section we examine various properties of the QP solver. We examine the ability to solve linear and non-linear ODEs, the ability to learn step sizes by differentiating through the step parameters and time complexity comparison with `torchdiffeq` used in neural network applications.

**Solving Linear and non-Linear ODEs.** We examine whether our quadratic programming solver is able to learn and solve linear ODEs accurately. We test solving second and third order linear ODEs with constant coefficients and compare with the RK4 solver from the `scipy` package. Results are shown in Figures 7 and 9 where we see a close match between the solver solutions.

Table 1: Solving 2d 85x85 Darcy flow Li et al. (2020c).

| Method | Loss |
|---|---|
| NNLi et al. (2020c) | 0.1716 |
| FCNLi et al. (2020c) | 0.0253 |
| PCANN Bhattacharya et al. (2021) | 0.0299 |
| RBMLi et al. (2020c) | 0.0244 |
| GNO Li et al. (2020a) | 0.0346 |
| LNO Li et al. (2020c) | 0.0520 |
| MGNOLi et al. (2020b) | 0.0416 |
| FNO Li et al. (2020c) | 0.0070 |
| MNN | **0.0065** |

**Learned Time Steps.** The differentiable QP solver can learn the discretization grid for learning and solving ODEs. The learned time steps can depend on solver progress and the grid can become finer in regions where the fit is poor. We show an example of this in Figure 8 for fitting a damped sinusoidal wave. Here we begin with a uniform grid and the steps become closer together in regions with bad fit.

**Learning and Complexity Comparison with RK4.** We performed an experiment to compare the QP solver against the RK4 solver (with the adjoint method) from the `torchdiffeq` package on a benchmark task. The task is to fit noisy sinuisoidal functions of varying lengths. We train for 100 iterations with dynamic Gaussian noise added at each iteration and use the MSE loss. The time for 100 iterations and the final loss is shown in Table 3 in the appendix. With 1000 steps, the *QP solver is about 200x faster than the RK4 solver,* and obtains a lower MSE loss. Visualizations of the learned functions can be seen in the appendix.

## 6 Conclusion

This paper presented Mechanistic Neural Networks (MNNs) – an approach for modeling complex dynamical and physical systems in terms of explicit governing mechanism. MNNs represent the evolution of complex dynamics in terms of families of differential equations. Any input or initial state can be used to compute a set of ODEs for that state using a learnable function. This makes MNNs flexible and able to model the dynamics of complex systems. The computational workhorse of MNNs is a new differentiable quadratic programming solver which allows a fast method for solving large batches of independent ODEs, allowing for efficient modeling of observable and hidden dynamics in complex systems. We have demonstrated the effectiveness of our method through experiments in diverse settings, showcasing its superiority over existing neural network approaches for modeling complex dynamical processes.

**Limitations and future work.** In this work, we have no way to measure or guarantee the identifiability of the computed equations, although in practice the computed equations might lie close to the true ones in terms of approximation capabilities. Inspired by the scientific method, it would also be interesting to explore applications of MNN in active setting, where experiments can be performed to falsify the predictions. Also, in the various experiments we did not explore the model design space much, better architectures and model choices can be made. We leave all above for future work.

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
