# A   OTHER DETAILS

**Linear programs.**   A linear program in the primal form is specified by a linear objective and a set of linear constraints.

$$
\begin{aligned}
\text{minimize} \quad & c^t x \\
\text{subject to} \quad & Ax = b, \\
& x \geq 0
\end{aligned}
\tag{12}
$$

where $A \in \mathbb{R}^{m \times n}$, $c \in \mathbb{R}^n$, $b \in \mathbb{R}^m$ the following specifies a linear program. Matrix $A$ and vector $b$ define the equality constraints that the solution for $x$ must comply with. $c^t x$ is a cost that the solution $x$ must minimize. The linear program can also be written in dual form,

$$
\begin{aligned}
\text{minimize} \quad & b^t \lambda \\
\text{subject to} \quad & A^t \lambda = c.
\end{aligned}
\tag{13}
$$

**Central Difference for Highest Order.**   The method proposed in Young (1961) does not add a smoothness constraint for the highest order derivative term. In cases where a more accurate highest order term is required, we also add a central difference constraint as a smoothness condition on the highest order term.

## A.1   ERROR ANALYSIS

We consider the case of a second order linear ODE with an $N$-step grid. For simplicity we consider a fixed step size $h$, i.e., $s_t = h$.

$$
c_2 u'' + c_1 u' + c_0 u = b,
\tag{14}
$$

Let $u(t)$ denote the true solution with initial conditions $u(0) = r$, $u'(0) = s$.

Define

$$
\tilde{u}_{t+1} = u_t + h u'_t + \frac{1}{2} h^2 u''_t,
\tag{15}
$$

$$
\tilde{u}'_{t+1} = h u'_t + \frac{1}{2} h^2 u''_t,
\tag{16}
$$

as Taylor approximations and $\tilde{u}''_t$ is obtained by plugging the approximate values in the ODE 14.

We consider the following Taylor constraints (expressions 8 9) for the function and its first derivative. We use the absolute-value error inequalities for conciseness, the case for equalities is similar.

$$
|\tilde{u}_{t+1} - u_{t+1}| \leq \epsilon
\tag{17}
$$

$$
|h \tilde{u}'_{t+1} - h u'_{t+1}| \leq \epsilon
\tag{18}
$$

**Step $t = 1$.**   From Taylor's theorem we have that for the first step, $t = 1$,

$$
u(h) = \tilde{u}_1 + O(h^3)
\tag{19}
$$

$$
u'(h) = \tilde{u}'_1 + O(h^2)
\tag{20}
$$

From 17, 18

$$
u_1 = \tilde{u}_1 + O(\epsilon + h^3)
\tag{21}
$$

$$
u'_1 = \tilde{u}'_1 + O(\epsilon/h + h^2)
\tag{22}
$$

This implies a local error at each step of $O(\epsilon + h^3)$ in $u_t$.

**Step $t = 2$.**   To estimate the error at step 2 we need to estimate the error in $u''_1$ at step 1.

For $u''_1$ we get the error by multiplying the error in $u_1$ by $\frac{c_0}{c_2}$ and that of $u'_1$ by $\frac{c_1}{c_2}$ and adding.

$$
u''_1 = \tilde{u}''_1 + O\left(\frac{c_1}{c_2}\left(\frac{\epsilon}{h} + h^2\right)\right) + O\left(\frac{c_0}{c_2}(\epsilon + h^3)\right)
\tag{23}
$$

Notice that $u_1''$ always appears with a coefficient of $h^2$. Assuming $\frac{c0}{c2}$ is $O(\frac{1}{h^2})$ and $\frac{c1}{c2}$ is $O(\frac{1}{h})$ we have

$$h^2 u_1'' = h^2 \tilde{u}_1'' + O(\epsilon + h^3) + O(\epsilon + h^3). \tag{24}$$

Each of the terms $u_1, h u_1', h^2 u_1''$ contribute an error of $O(\epsilon + h^3)$ to $u_2$ plus an additional error of $O(h^3)$ arising from the Taylor approximation and an error of $\epsilon$ arising from the inequalities 17, 18.

$N$ **Steps.** Proceeding similarly, after $N$ steps we get a cumulative error of $O(N(\epsilon + h^3))$.

For $\epsilon \approx h^3$ and $N \approx 1/h$, we get an error of $O(h^2)$ for $N$-steps under the assumption that $\frac{c0}{c2}$ is $O(\frac{1}{h^2})$ and $\frac{c1}{c2}$ is $O(\frac{1}{h})$.

The analysis implies that the cumulative error can become large for equations where $\frac{c0}{c2}, \frac{c1}{c2}$ are large. Or, in little omega notation, $\frac{c0}{c2}$ is $\omega(\frac{1}{h^2})$ and $\frac{c1}{c2}$ is $\omega(\frac{1}{h})$ .

## A.2 NON-LINEAR ODE DETAILS

In this section we give some further details regarding the formualtion of non-linear ODEs. We illustrate with the following non-linear ODE as an example

$$c_2(t)u'' + c_1(t)u' + d_0(t)u^3 + d_1(t)u'^2 = b, \tag{25}$$

where $u^3$ and $u'^2$ are non-linear functions of $u, u'$.

As described in Section 3.1, we create one set of variables $u_t$ for each time step $t$ for the solution $u$. In addition we create a set of variables $\nu_{0,t}$ for $u^3$ and another set of variables $\nu_{1,t}$ for $u'^2$. In addition we create variables $\nu_{i,t}', \nu_{i,t}''$ for derivatives for each $i$, as in Section 3.1.

Next we build constraints. We add equation constraints for each time step as follows.

$$c_{2,t}u_t'' + c_{1,t}u_t' + d_{0,t}\nu_{0,t} + d_{1,t}\nu_{1,t} = b_t, \forall t \in \{1, \ldots, n\}. \tag{26}$$

We add smoothness constraints for each $\nu_{i,t}$ in the same way as described for $u_t$ in expressions 8, 9.

Next we solve the quadratic program to obtain $u_t, u_t', \nu_{0,t}, \nu_{1,t}$ in the solution. Now we need to relate the $nu_{0,t}, nu_{1,t}$ variables to non-linear functions of the $u_t, u_t'$ variables. For this we add the term to the loss function

$$\frac{1}{N} \sum_t (u_t^3 - \nu_{0,t})^2 + (u_t'^2 - \nu_{1,t})^2.$$

Figure 9 shows solving and fitting of a non-linear ODE.

## A.3 DISCOVERY

We provide further details of the discovery method from Section 5.1. This method follows the SINDy Brunton et al. (2016) approach for discovering sparse differential equations using a library of basis functions. Unlike SINDy, which resorts to linear regression, the MNN method uses deep neural networks and builds a non-linear model which allows modeling of a greater class of ODEs.

The method requires a set of basis functions such as the polynomial basis functions up to some maximum degree. Over two variables $x, y$ this is the set of functions $\{0, x, y, x^2, xy, y^2, xy^2, \ldots, y^d\}$ for some maximum degree $d$. Let $k$ denote the total number of basis functions.

Next we are given some observations $X = [(x_0, y_0), (x_1, y_1), \ldots, (x_{n-1}, y_{n-1})]$ for $n$ steps. We first transform the sequence by applying an MLP to the flattened observations producing another sequence of the same shape.

$$\tilde{X} = [(\tilde{x}_0, \tilde{y}_0), \ldots, (\tilde{x}_{n-1}, \tilde{y}_{n-1})] = \text{MLP}(X)$$

We apply the basis functions to $\tilde{X}$ to build the basis matrix $\Theta \in \mathbb{R}^{n \times k}$.

$$\Theta(\tilde{X}) = \begin{bmatrix} 1 & \tilde{x}_0 & \tilde{y}_0 & \tilde{x}_0^2 & \tilde{x}_0\tilde{y}_0 & \tilde{y}_0^2 & \cdots \\ 1 & \tilde{x}_1 & \tilde{y}_1 & \tilde{x}_1^2 & \tilde{x}_1\tilde{y}_1 & \tilde{y}_1^2 & \cdots \\ \vdots & \vdots & \vdots & \vdots & \vdots & \vdots & \\ 1 & \tilde{x}_{n-1} & \tilde{y}_{n-1} & \tilde{x}_{n-1}^2 & \tilde{x}_{n-1}\tilde{y}_{n-1} & \tilde{y}_{n-1}^2 & \cdots \end{bmatrix} \tag{27}$$

Let $\xi \in \mathbb{R}^{n \times 2}$ be a set of parameters, with each column specifying the active basis functions for the corresponding variable in $[\dot{x}, \dot{y}]$.

The ODE to be discovered is then modeled as

$$[\dot{x}, \dot{y}] = f(\Theta(\tilde{X})\xi) \tag{28}$$

where $f$ is some arbitrary differentiable function. Note that for SINDy $\tilde{X} = X$ and $f$ is the identity function and the problem is reduced to a form of linear regression adapted to promote sparsity in $\xi$. SINDy estimates the derivatives using finite differences with some smoothing methods.

With MNN the ODE 28 is solved using the quadratic programming ODE solver to obtain the solution $\bar{x}_t, \bar{y}_t$ for $t \in \{0, \ldots, n-1\}$. The loss is then computed as the MSE loss between $\tilde{x}_t, \tilde{y}_t, \bar{x}_t, \bar{y}_t$ and the data $x_t, y_t$.

$$\text{loss} = \frac{1}{N} \sum_t (\tilde{x}_t - x_t)^2 + (\tilde{y}_t - y_t)^2 + (\bar{x}_t - x_t)^2 + (\bar{y}_t - y_t)^2$$

# B    FURTHER EXPERIMENTS

## B.1    QUANTITATIVE COMPARISON WITH NEURAL ODE ON THE N-BODY PROBLEM

We report quantitative results for a comparison with Neural ODE on the n-body prediction problem for 2 and 10 bodies. For Neural ODE we used a neural network with 512 units and one hidden layer and use the `dopri5` method for solving ODEs and train for 50 epochs. The results are report in Table 2. We find Neural ODEs to be a poor model for these datasets with significant underfitting whereas MNNs can fit the dataset easily with good evaluation score.

Table 2: N-body problem. Comparing with Neural ODE.

| Method | Training Loss | Eval Loss |
|---|---|---|
| 2-body | | |
| Neural ODE | 38.5 | 321.0 |
| MNN | 0.035 | 0.114 |
| 10-body | | |
| Neural ODE | 165.0 | 201.0 |
| MNN | 0.0934 | 0.853 |

## B.2    VALIDATING THE QUADRATIC PROGRAMMING ODE SOLVER

First we examine whether our quadratic programming solver is able to solve linear ODEs accurately. For simplicity we choose the following second and third order linear ODEs with constant coefficients.

$$u'' + u = 0 \tag{29}$$

$$u''' + u'' + u' = 0 \tag{30}$$

For the QP solver we discretize the time axis into 100 steps with a step size of 0.1. We compare against the ODE solver `odeint` included with the SciPy library. The results are shown in 7 where we show the solutions, $u(t)$, for the two ODEs along with the first and second derivatives, $u'(t), u''(t)$. The results from the two solvers are almost identical validating the quadratic programming solver.

Next we examine the ability of the solver to learn the discretization. We learn an ODE to model a damped sine wave where each step size is a learanable parameter initial to 0.1 and modeled as a sigmoid function. We show the results in Figure 8 for a sample of training steps. We see the step sizes varying with training and the steps generally clustered together in regions with poorer fit.

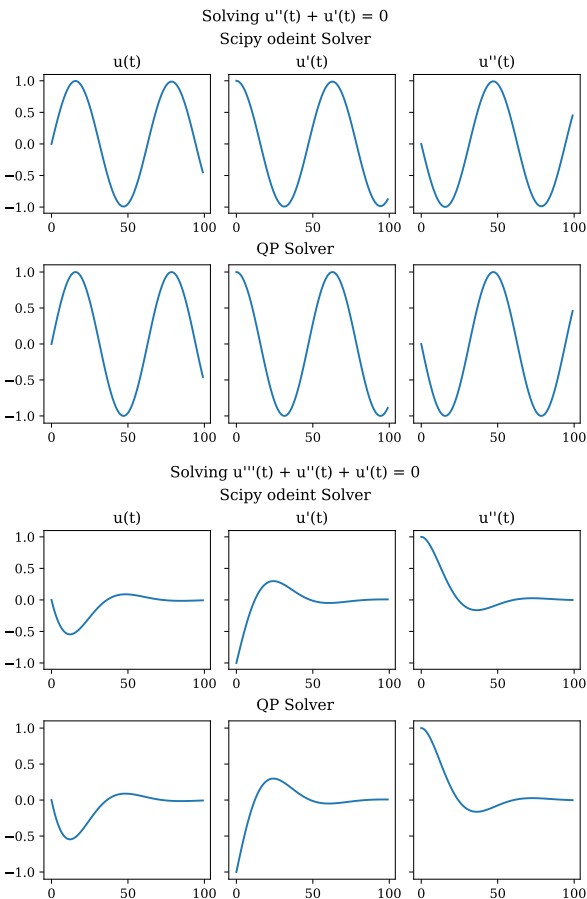

Figure 7: Comparing ODE solvers on 2nd and 3rd order ODEs.

Next we demonstrate a non-linear equation. For this we introduce a variable in the QP solver for a non-linear term add a squared loss term as described in the paper. We use the equation $c_2(t)y'' + c_1(t)y' + c_0(t)y + \phi(t)y^2 = 1$, with time varying coefficients and fit a sine wave. The result is in Figure 9. The ODE fits the sine wave and at the same time the non-linear solver term fits the true non-linear function of the solution.

### B.3 LEARNING WITH NOISY DATA

We perform an simple experiment illustrate how the ODE learning method can fit ODEs to noisy data. We generate a sine wave with dynamic Gaussian noise added during each training step. We train two models: the first a homogeneous second order ODE with arbitrary coefficients and the second a homogeneous second order ODE with constant coefficients. We also train a model without noise. The results are shown in Figure 10. The figures show that the method can learn an ODE in the presence of noise giving a smooth solution. The model with constant coefficients learns the following ODE.

$$0.92023u'' - 0.00016u' + 0.228u = 0,$$

with (learned) initial conditions $u(0) = -0.031799$ and $u'(0) = 2.3657$.

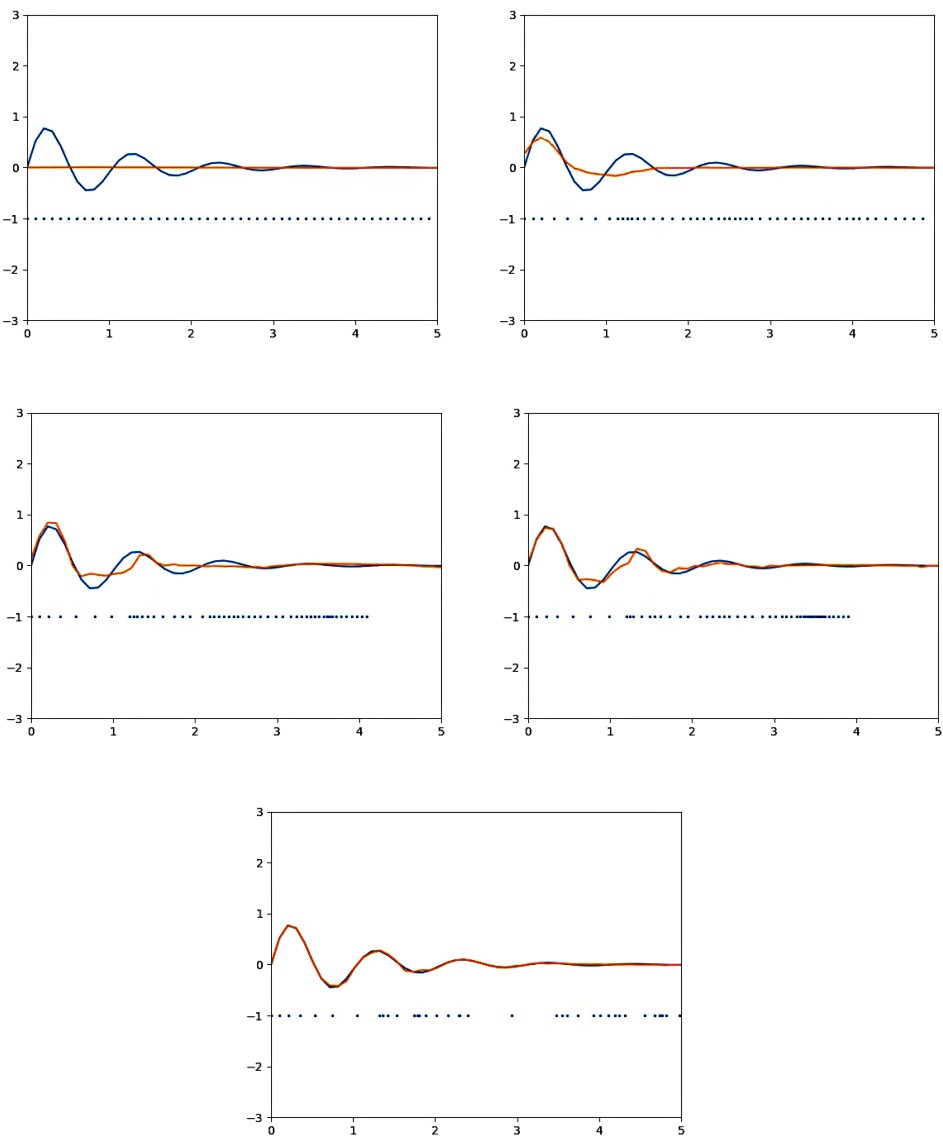

Figure 8: Demonstrating a learned grid for fitting a damped sinuoidal wave (blue curve) over the course of training. The dots show the learned grid positions. The grid generally becomes finer for regions where the fit is poorer.

## C  EXPERIMENTAL DETAILS

### C.1  DISCOVERY OF GOVERNING EQUATIONS

#### C.1.1  DISCOVERING GOVERNING EQUATIONS OF SYSTEMS WITH RATIONAL FUNCTION DERIVATIVES

In Figure 12 we plot the vector fields learned with SINDy and with MNNs. MNNs are considerably more accurate.

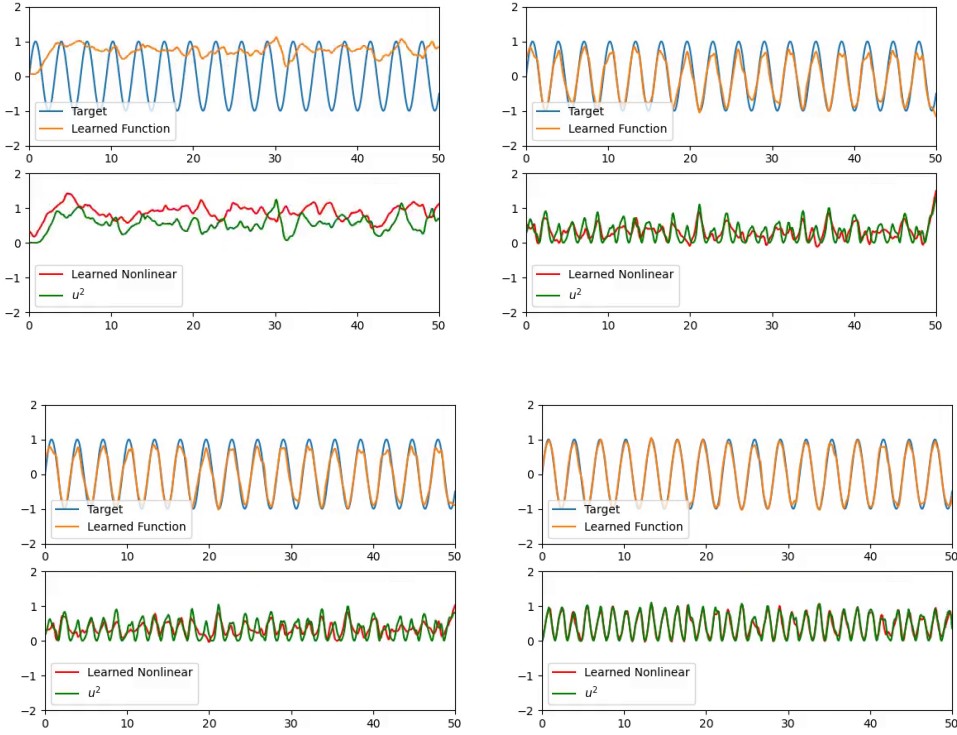

Figure 9: Demonstrating fitting a sine wave with a non-linear ODE $c_2(t)y'' + c_1(t)y' + c_0(t)y + \phi(t)y^2 = 1$. The non-linear function is $y^2$ and the bottom shows the solver variable fitting the non-linear function.

### C.1.2 DISCOVERED EQUATIONS.

**MNN Lorenz**

$$x' = -10.0003x + 10.0003y$$
$$y' = 27.9760x + -0.9934y - 0.9996xz$$
$$z' = -2.6660z + 0.9995xy$$

**SINDy Lorenz**

$$x' = -10.000x + 10.000y$$
$$y' = 27.998x + -1.000y + -1.000xz$$
$$z' = -2.667z + 1.000xy$$

**MNN Non-linear**

$$x' = \tanh(-0.7314x + 0.5545y + -1.2524x^2 + -0.1511xy + 0.2134y^2)$$
$$y' = \tanh(0.9879x + 1.0005y + 0.1742x^2)$$

**SINDy Non-linear**

$$x' = -1.968x + 0.985y + -0.054x^2$$
$$y' = 1.466y + 11.892x^2 + -5.994xy + 0.085y^2$$

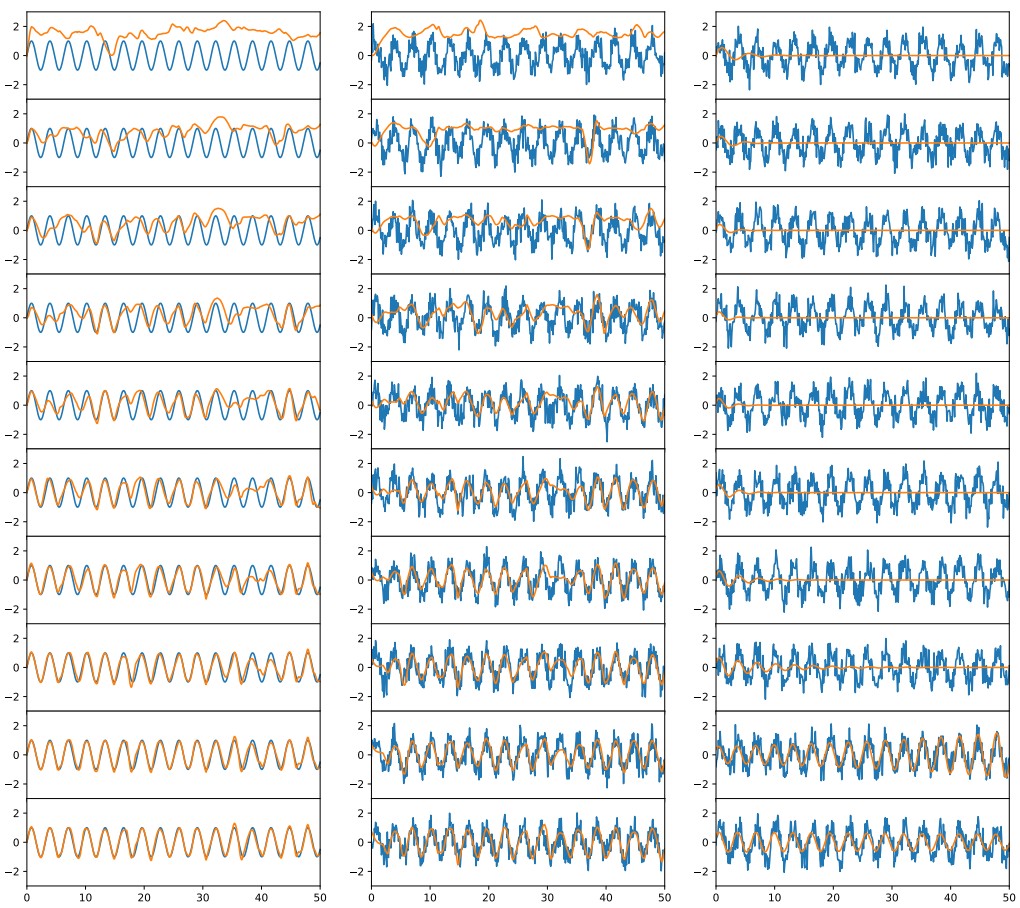

Figure 10: Learning sine waves without and with dynamically added Gaussian noise with 2nd order ODE with arbitrary coefficients (middle) and constant coefficients (right). The figure on the right corresponds to the ODE $0.92023u'' - 0.00016u' + 0.228u = 0$.

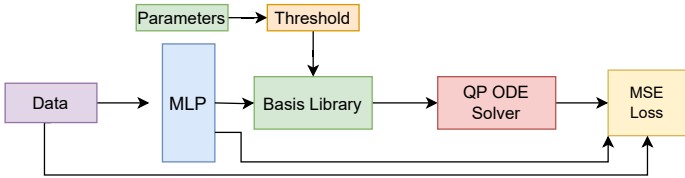

Figure 11: Governing equation discovery architecture

**MNN Rational**

$$x' = \frac{-0.9287x + 0.4386y + -1.1681x^2 + 0.3545y^2}{0.4871 + 0.8123x + 0.0984x^2 + 0.3700xy + 0.3081x^2}$$

$$y' = \frac{0.6360x + 0.5971y + 0.3267x^2}{0.6090 + 0.7507x^2 + 0.5694y^2}$$

**SINDy Rational**

$$x' = -1.705x + 0.899y + -0.318x^2$$

$$y' = -0.795 + 3.072y + 4.777x^2 + 6.892xy + -4.681y^2$$

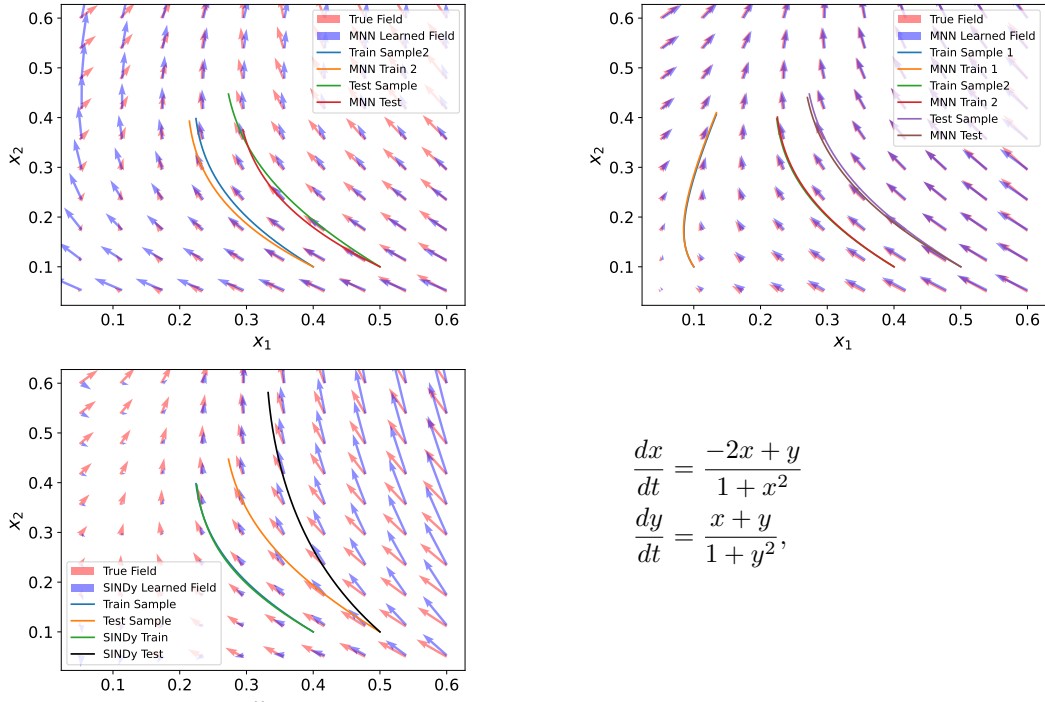

$$\frac{dx}{dt} = \frac{-2x + y}{1 + x^2}$$
$$\frac{dy}{dt} = \frac{x + y}{1 + y^2},$$

Figure 12: Learned ODE vector fields for MNN and SINDy with rational function derivatives and one and two training trajectories. MNN can handle multiple input examples. The ground truth ODE is also shown.

## C.2 Nested Circles

For the nested circles (Section 5.2) experiment we use a second order ODE with coefficients computed with a single layer and the right hand side is set to 0. We use a step size of 0.1 and length 30. However, as we note, 5 time steps are enough for accurate classification. The loss function is the cross entropy loss.

MNNs obtain an explicit linear ODE per datapoint that governs the evolution of the point. The example we give is for one of the ODEs for one point and for a 5-time step evolution. This computed equation is sufficient for perfect classification.

## C.3 Airplane Vibrations

For this experiment (Section 5.3) we use an MNN with a second order ODE, step size of 0.1 and 200 steps during training. The coefficients and constant terms are computed with MLPs with 1024 hidden units.

## C.4 2-Body Problem

We give experimental detail for the experiment from Section 5.4.

**Learning Trajectory.** For learning the 2-body trajectory we use an MNN with second order ODEs. The time axis is discretized into steps of size 0.01 and length 50. The input $x$ is the initial positions and velocities of the two objects. The coefficients of the ODE are computed by an MLP with one hidden layer and 2048 units given the pairwise distance and velocities as input. The force is computed by an MLP with two hidden layers with 1024 units and we use Newton's 3rd law for the complementary force. Solving the ODE gives the position, velocity and acceleration over 50 time steps. The loss is the MSE loss for position and velocity.

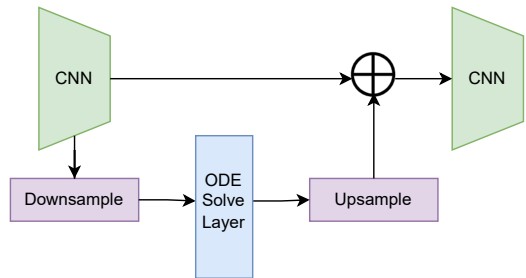

Figure 13: PDE module architecture

**Discovering Mass and Force Parameters.** For this part of the experiment we use an MNN with a restricted ODE to match Newton's second law. In the MNN model for this experiment, we use the same coefficient for the second derivative term for all time steps with the remaining coefficients fixed to 0, that is $c_2(t) = c$ and $c_1(t) = 0, c_0(t) = 0$. $b(t) = F_t$ corresponds to the force term which is computed by a neural network from the initial position and velocity with two hidden layers of 1024 units and Newton's second law $F_{21} = -F_{12}$. We use a step size of $0.01$ and run for 50 time steps. The loss is the MSE loss for position and velocity.

## C.5  N-BODY PROBLEM

The $n$-body setup is similar to the trajectory learning setup for the 2-body problem. The difference is that the force neural network is used compute pairwise forces between the objects using Newton's third law and the superposition principle so that the individual forces on an are added to produce the final force.

## C.6  PDE SOLVING

In Figure 13 we show the MNN architecture we used to solve PDEs. We use the 2d Darcy Flow dataset used by Li et al. (2020c) scaled to 85x85.

## C.7  COMPARING RK4 WITH THE QP SOLVER

Table 3: Comparing the QP solver with the RK4 solver with a step size of 0.1 on fitting noisy sinusoidal waves of 300 and 1000 steps. Showing MSE loss and time.

| Steps | QP (seconds) | RK4 (seconds) | QP Loss | RK4 Loss |
|-------|--------------|---------------|---------|----------|
| 40    | 1.52         | 28.06         | 11.4    | 29.3     |
| 100   | 1.61         | 64.57         | 27.9    | 35.6     |
| 300   | 1.76         | 211.52        | 52      | 96.8     |
| 500   | 2.12         | 359.7         | 128     | 301      |
| 1000  | 3.68         | 666.69        | 292     | 589      |

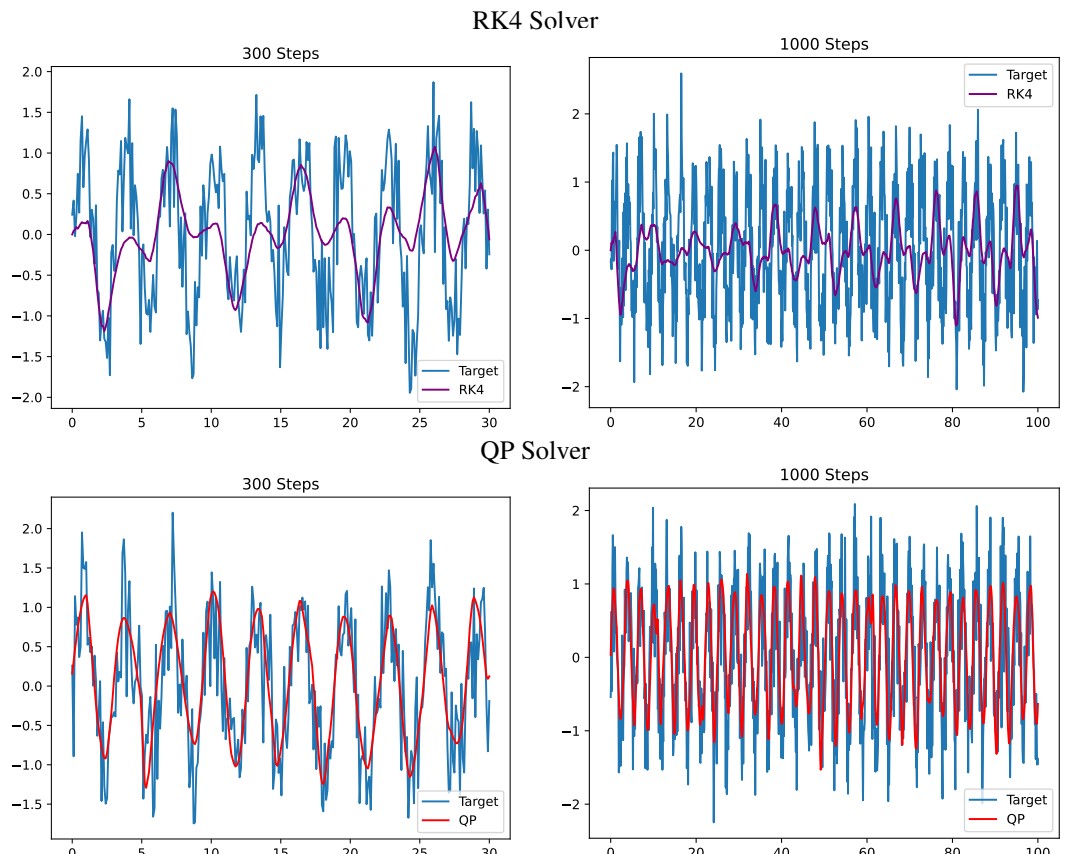

Figure 14: Comparison of RK4 solver from *torchdiffeq* and our QP solver for fitting sinusoidal waves with Gaussian noise added at each iteration. Length of the wave and number of steps is 300 (left column) and 1000 (right column). Step size is 0.1. Trained for 100 iterations. The QP solver has better performance (and efficiency) for longer trajectories.

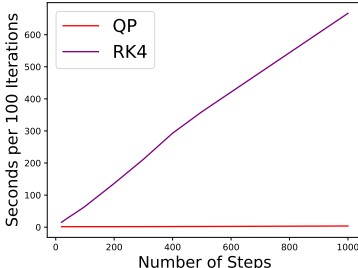

Figure 15: Number of seconds per 100 iterations for fitting noisy sinusoidal waves. The QP solver is significantly more efficient over longer times due to its parallelism.