# OpenReview forum: "Mechanistic Neural Networks"
_ICLR.cc/2024/Conference — Submitted to ICLR 2024_

### Official Review · Reviewer_L7km · 2023-10-30

**Soundness:** 3 good
**Presentation:** 1 poor
**Contribution:** 2 fair
**Rating:** 3
**Confidence:** 2

**Summary:**

This paper proposes mechanistic neural networks, a framework to learn differential equations. Given an input, these networks output (the parameters of) a differential equation. To solve the differential equations and backprogate through the solution, the authors first leverage old work from the 1960s, which frames numerically solving ODEs as a quadratic optimization problem. Then, they leverage more recent work allowing to solve and backpropagate through the solution of quadratic optimization problems given by neural networks. This way of solving and backpropagating through ODEs is more amenable to parallelization in a GPU than other methods used in neural ODEs. Finally, the authors perform a large amount of qualitative comparisons to show the effectiveness of their method.

Before starting my review, I do want to highlight that I am not an expert in differential equations nor in neural ODEs. While I found the high-level idea of the paper to be understandable, the details were very confusing. Notation is poorly chosen and the paper reads like a rushed submission. This opinion might nonetheless be a consequence of my own lack of expertise in the topic, so I will happily change my opinion on this if other reviewers who are more knowledgeable in the area disagree.

**Strengths:**

1. I believe the authors are tackling a highly relevant problem, which can be of interest not only to the machine learning community itself, but to physicists and scientists in general who are interested in using machine learning.

2. Improving the computational efficiency of ODE solvers with neural networks in mind is also a relevant problem, and the proposed method seems to perform well.

**Weaknesses:**

3. The notation is quite confusing, and I think the authors use the same symbols to denote different objects in different sections. For example, in section 2, $x$ is the input to the neural network, and $c$, $g$, and $b$ define the differential equation (eq 2). In section 3, $x$ is the variable being optimized over, $c$ is part of the objective, $b$ provides a constraint, and the matrix $G$ is a multiple of the identity that has nothing to do with the $g$ scalars defined in the previous section. The notation used by the authors is highly suggestive of a connection between $x$, $b$, $c$, $g$, and $G$ in these two sections, but unless I hugely misunderstood, there is no connection other than the quantities in section 2 characterize the optimization problem in eq 5.

4. I also found eqs 8 and 9 to not be clear, could you please further explain their role and how they are obtained?

5. The paper emphasizes throughout how it is fundamentally different than neural ODEs, making it sound like neural ODEs cannot be used to solve tasks that mechanistic neural networks can be used for. Yet, most of the empirical comparisons are carried out against neural ODEs.

5. Other than table 1, which compares exclusively solvers, the paper presents no quantitative comparisons against its baselines, only plots qualitatively comparing trajectories.

Finally, I have many minor comments:

- The line after eq 2 has $u^{(k)}$, should this be $u^{(i)}$?

- The line after eq 3 says $c_{d, t}$ and $c_d$, should these be $c_{i, t}$ and $c_i$, respectively?

- Using the same symbol for transpose than for time indexing is also confusing, I'd recommend using ^\top.

- \citep and \citet are used exchangeably throughout the manuscript, please use each one when appropriate only.

- In the abstract, lists are numbered as (i), (ii); whereas throughout the manuscript as (1), (2).

- Missing parenthesis above eq 4.

- Bottom of page 6: "not enough represent" -> "not enough to represent".

- "Planar and Lorenz System" paragraph: "represent" -> "represented"

- The phrasing "where we must classify per particles which class out of two it belongs too" is awkward.

- "a wing on gives rise" -> "a wing gives rise"

- "only and to predicting" -> "only and to predict"

- Period missing after the first sentence in sec 5.4.

**Questions:**

What is $\hat{r}$ in 5.4?

---

> ### Author Response · Authors · 2023-11-14
>
> We thank the reviewer for the comments.
>
> >> I also found eqs 8 and 9 to not be clear, could you please further explain their role and how they are obtained?
>
> **Smoothness Constraint Elaboration**. The purpose of the constraints in eqs 8 and 9 is to enforce the solution of the ODE to be smooth.
>
> The QP defines ODE solution as variables $u\_t, u'\_t, u''\_t$ over the time grid.
> However, these would not be smooth without additional constraints.
>
> We use Taylor approximations to enforce smoothness between solution at time $t$ and $t+1$.
>
> Given $u\_t, u'\_t, u''\_t$ we build the second order Taylor polynomial to approximate the function at time $t+1$ with step size $h$ as
> $$\tilde{u}\_{t+1} = u\_t + hu'\_t + \frac{1}{2}h^2 u''\_t. $$
> Next we say that the difference between the approximation and the solution at $t+1$ should be $\epsilon$.
> $$\tilde{u}\_{t+1} - u\_{t+1} = \epsilon,$$
> which is the first equation in 8 which enforces the function to be smooth by minimizing $\epsilon^2$ which is the optimization objective.
>
> Similarly, the next equation in 8 enforces the first derivative to be smooth using the Taylor approximation.
>
> The constraints in 8 say that we should be able to approximate the next time step $t+1$ using values at $t$.
> The constraints in 9 go backward using values at $t+1$ to approximate values at $t$.
> We find both constraints to be necessary to get proper solutions.
>
>
> >>The paper emphasizes throughout how it is fundamentally different than neural ODEs, making it sound like neural ODEs cannot be used to solve tasks that mechanistic neural networks can be used for. Yet, most of the empirical comparisons are carried out against neural ODEs.
>
> **Neural ODE Comparison**. We emphasize the difference of our _approach_ in combining Neural networks and ODEs which is very different from that of Neural ODEs.
> Nevertheless, there is an overlap in the types of tasks for which Neural ODEs and MNN can be used, such as modeling dynamical systems and processes.
> We compare against Neural ODE for such tasks (Classification, Vibration prediction, N-body prediction) in Sections 5.2,5.3,5.5.
> MNNs have better performance on these tasks because the underlying solver performs better at learning dynamical processes than the solver type used in Neural ODE (see Section C.7).
>
> On the other hand, MNNs are also designed for discovery. In discovery experiments we compare against a SINDy (Section 5.1) which is a popular sparse discovery method for ODEs.
>
> For our PDE solving architecture we compare against FNO-based and similar methods.
>
> >> Other than table 1, which compares exclusively solvers, the paper presents no quantitative comparisons against its baselines, only plots qualitatively comparing trajectories.
>
> **Quantitative Results**. We point out that there are several other quantitative comparisons in the paper.
>
> In Figure 4, we plot loss curves (left) and prediction error (second from left) which are quantitative and compare against Augmented Neural ODE and Second Order Neural ODEs.
>
> In Table 2 in the appendix we compare the training and evaluation results on the 2-body and 10-body problem against Neural ODE.
>
> In Table 3 in the appendix we compare the QP solver against the RK4 solver from torchdiffeq, where we look at the ability of the two solvers to fit long sinusoidal waves (300 and 1000 steps) and their respective training times.
>
> Finally in Table 1 (Section 5.5) we give quantitative comparisons for a PDE solving architecture against other well-known deep learning based PDE solving architectures.
> These methods are not solvers in the traditional sense of differential equations solving but are learning-based approaches.
>
> The above quantitative results are in addition to the host of results that we have shown for discovering governing equations (Sections 5.1, 5.4, C.1.2) and prediction (Sections 5.2,5.3,5.5).
>
> **Notation**.
> >>The notation is quite confusing, and I think the authors use the same symbols to denote different objects in different sections ...
>
> Unfortunately, describing our approach has to combine methods from differential equations, optimization and deep learning where each has standard notation and some overlap has been difficult to avoid.
> We have made many of the changes suggested and appreciate the reviewer's suggestions for improvement.
>
> >> What is $\hat{r}$ in 5.4?
>
> $\hat{r}$ is the radius unit vector that gives the direction of force in Newton's law of gravitation.
> We have added a clarification.
>
> Finally we appreciate the typographical corrections pointed out by the reviewer and are grateful for the review in general.

---

> > ### Comment · Reviewer_L7km · 2023-11-21
> >
> > I thank the authors for their reply and the updated draft. While some of the misunderstandings I had have been clarified, I still find that the exposition can be made clearer, a point that is also echoed by reviewer 5iBu. I will thus keep my score as is.

---

### Official Review · Reviewer_5iBu · 2023-10-31

**Soundness:** 2 fair
**Presentation:** 1 poor
**Contribution:** 2 fair
**Rating:** 3
**Confidence:** 4

**Summary:**

The paper introduces Mechanistic Neural Networks (MNNs), a neural module that represents the evolution of its input data in the form of differential explicit equations. Unlike some traditional neural networks that return vector-valued outputs, MNNs output the parameters of a mechanism in the form of an explicit symbolic ordinary differential equation. MNNs employ a new parallel and differentiable ODE solver design that can solve large batches of independent ODEs in parallel on GPU.

**Strengths:**

The introduction of Mechanistic Neural Networks provides a new approach to learning differential equations from the evolution of data.

**Weaknesses:**

1. The paper's clarity is wanting. For example, around eq(2), there's an inconsistency in notation with both $c_i(t;x)$ and $c_i(t)$ being used. Which of these is the intended notation? Additionally, there's no explicit description or definition of $b(t;x)$.
2. On page 2, the statement "In general, one ODE is generated for a single data example $x$ and a different example $x'$ would lead to a different ODE" is made. Could the authors elucidate why this is a characteristic of the modeling approach presented in eq(2)?
3. If a new instance $x'$ necessitates retraining the model, wouldn't it be more streamlined to directly learn a neural ODE through parameter optimization, bypassing the need for coefficients as functions of $x$?
4. The paper's approach to solving any nonlinear ODEs using equality-constrained quadratic programming must have gaps. These gaps aren't clearly addressed. Relying on such an algorithm to solve any ODE without theoretical guarantees is precarious. A more transparent discussion on potential limitations is needed.
5. The methodology for handling nonlinear ODEs, as presented on page 4, lacks clarity and could benefit from a more detailed exposition.
6. The literature review in section 4 seems outdated, with the most recent references dating back to 2020. A comprehensive literature survey, including more recent and relevant baselines, would strengthen the paper's context. For example for the ODE modelling, [1][2] may be included.
7. The term $\Theta \xi$ on page 6 is introduced without clear definition or context. Could the authors provide clarity on this?
8. The paper delineates two primary components: the learning of the ODE and its subsequent solving. However, the experimental section seems to lack comprehensive ablation studies that convincingly demonstrate the efficacy of each individual component.

[1] Kidger, Patrick, et al. "Neural controlled differential equations for irregular time series." Advances in Neural Information Processing Systems 33 (2020): 6696-6707.

[2] Morrill, James, et al. "Neural rough differential equations for long time series." International Conference on Machine Learning. PMLR, 2021.

**Questions:**

Please clarify the issues raised in the weaknesses section.

---

> ### Author Response · Authors · 2023-11-14
> **Response 1/2**
>
> We thank and appreciate the reviewer's comments and suggestions.
>
> >>The paper's clarity is wanting. For example, around eq(2), there's an inconsistency in notation with both $c\_i(t;x)$ and  $c\_i(t)$ being used. Which of these is the intended notation? Additionally, there's no explicit description or definition of $b(t;x)$ .
>
> The notation $c\_i(t;x)$ was intended to make dependence of the ODE coefficients on neural network input $x$ explicit.
> For a fixed datum $x$, we use $c\_i(t)$.
> To avoid confusion, we have made the notation consistent in the updated draft and have given a definition of $b(x;t)$, defined similarly to the other coefficients.
>
> >> On page 2, the statement "In general, one ODE is generated for a single data example and a different example would lead to a different ODE" is made. Could the authors elucidate why this is a characteristic of the modeling approach presented in eq(2)?
>
> **Per-instance ODEs**. The ODE in equation 2 depending on the input datum $x$ means that we have **_one ODE per input datum_** in the dataset in the most general case. For instance, if wanted the MNN can derive a different governing ODE for each new input sequence trajectory, initial condition, etc. Of course, if wanted the MNN can also be defined to derive a *single governing ODE* from all the data in the dataset, e.g when there is a single governing equation in the discovery experiments in Sections 5.1 and 5.4.
>
> This draws a contrast with Neural ODEs, which are only able to model a global ODE for an entire dataset.
>
> >> If a new instance $x'$ necessitates retraining the model, wouldn't it be more streamlined to directly learn a neural ODE through parameter optimization, bypassing the need for coefficients as functions of $x$ ?
>
> **Retraining**. It is **_not correct that a new instance $x'$ requires retraining the model_**.
> In our model a neural network produces the parameters $c\_i, b, \phi\_k$ (equation 2) of an ODE for a given input $x$.
> A dataset can then be modeled by a family of ODEs by training over the dataset.
> As we mention above, MNN is more general than neural ODE and is able to model per-instance ODEs without retraining, simply returning the datum-specific coefficients by conditioning on the datum.
> This is indicated by the dependence of the produced ODE on the input datum $x$.
>
> >> The paper's approach to solving any nonlinear ODEs using equality-constrained quadratic programming must have gaps. These gaps aren't clearly addressed. Relying on such an algorithm to solve any ODE without theoretical guarantees is precarious. A more transparent discussion on potential limitations is needed.
>
> **Theoretical Analysis**. We have added an error analysis section in the updated draft in appendix A.1 for linear second order ODEs with the QP solver
> $$c\_2 u'' + c\_1 u' + c\_0 u = b.$$
>
> **_Sketch_**. The analysis proceeds by estimating total error introduced by the error in the second order Taylor approximation $O(h^3)$ and the error from the constraint $\epsilon$.
> In the first step the function and derivative values are known from the initial value constraints leading to an error of $O(\epsilon + h^3).$
> For the next step we use the approximate function and derivative values, plugged into the ODE, and show that the total error over $N$ steps remains bounded by $O(N(\epsilon + h^3))$, under the assumption that $\frac{c\_1}{c\_2}$ and $\frac{c\_0}{c\_2}$ are bounded by $O(\frac{1}{h^2})$ and $O(\frac{1}{h}) $, respectively.
>
> The analysis shows that the method has a local error of $O(\epsilon + h^3)$, where $\epsilon$ is the QP error variable and $h$ is a fixed step size.
> For an $N$ step grid, the analysis shows that this error becomes
> $O(N(\epsilon + h^3))$.
>
> In particular for $\epsilon = h^3$, which is reasonable in practice, and  $N=1/h$ the global error simplifies to $O(h^2)$.
> This error depends on the number of terms of the Taylor expansion and could be further reduced by using central difference approximations for all derivatives, investigation of which we leave to future work.
>
> The analysis implies that the cumulative error becomes potentially large only when $\frac{c0}{c2}$, $\frac{c1}{c2}$ are very large:
> In little omega notation, when $\frac{c0}{c2}$ is $\omega(\frac{1}{h^2})$ and $\frac{c1}{c2}$ is $\omega(\frac{1}{h})$.

---

> ### Author Response · Authors · 2023-11-14
> **Response 2/2**
>
> >>The methodology for handling nonlinear ODEs, as presented on page 4, lacks clarity and could benefit from a more detailed exposition.
>
> **Non-linear ODEs**. We have added further clarification in the appendix A.2 with an example non-linear ODE.
>
> Briefly, to solve an equation like
> \begin{equation}
> c\_2(t) u'' + c\_1(t) u' + d(t) u^3 = b,
> \end{equation}
> where $u^3$ is non-linear functions of $u$, we solve the following two equations.
> \begin{align}
> c\_2(t) u'' + c\_1(t) u' + d(t) \nu= b\\\\
> u^3 = \nu
> \end{align}
>
> We solve the first equation by the QP solver by creating another set of variables $\nu\_t$ in addition to $u\_t$.
> We add equation constraints that include both types of variables and include smoothness constraints for $\nu\_t$.
>
> Solving the program gives a solution for $u\_t$ and $\nu\_t$.
> We enforce the non-linearity in the neural network loss function by add an MSE term $\sum_t (u\_t^3 - \nu\_t)^2$ which imposes that constraint that $u^3 = \nu$.
> This shows how non-linear relationships in the ODE terms are learned.
>
>
> >> The literature review in section 4 seems outdated, with the most recent references dating back to 2020. A comprehensive literature survey, including more recent and relevant baselines, would strengthen the paper's context. For example for the ODE modelling, [1][2] may be included.
>
> **References**. Thank you for the suggestion about more recent references; we will consider this for the final draft.
> This is partly due to the fact that there hasn't been much significant development in Neural ODE style modeling over the past few years.
> The reviewer's suggested references are also from 2020 and 2021.
>
> We demonstrate that our method overcomes significant limitations of the Neural ODE framework and adds directions such as explicit equation discovery which were not possible in the Neural ODE method, in addition to faster parallel solving and better learning.
>
> >> The term $\Theta \xi$ on page 6 is introduced without clear definition or context. Could the authors provide clarity on this?
>
> **SINDy Notation**. In Section 5.1, we assume and follow the notation used by Brunton et al (2016) that introduces the SINDy method.
> For clarity and completeness we have included further details in the appendix A.3 with a reference in the main text.
>
> Briefly $\Theta$ is a data matrix that applies polynomial basis functions to time-series input $x\_0, x\_1 \ldots x\_t$.
> \begin{equation}
> \Theta =
> \begin{bmatrix}
> 1 & x\_0 & x\_0^2 & \ldots\\\\
> 1 & x\_1 & x\_1^2 &\ldots\\\\
> \vdots & \vdots & \vdots \\\\
> 1 & x\_t & x\_t^2 &\ldots
> \end{bmatrix}
> \end{equation}
>
> $\xi$ is a sparse vector that selects a small number of basis functions. So ODE like
> $$ x'  = x + x^2$$
> may be represented with the appropriate choice of $\xi$ as $ x'  = \Theta\xi$.
>
> >>The paper delineates two primary components: the learning of the ODE and its subsequent solving. However, the experimental section seems to lack comprehensive ablation studies that convincingly demonstrate the efficacy of each individual component.
>
> Our original submission included the following ablations in the appendix (referenced in Section 5.7) due to lack of space.
>
> **RK4 Solver Comparison**. In Section B.2 we validate the QP solver and compare the solution with the RK4 solver solution with the SciPy Python package (Figure 7).
>
> **Non-linear ODEs**. We show learning a non-linear ODE also in B.2 and Figure 9.
>
> **Step Learning**. In Section B.2 and Figure 8 we show that the method learns step sizes during training.
>
> **Learning with Noise**. In Section B.3 we show that the method is able to learn ODEs from noisy data both with time-dependent and time-independent coefficients.
>
> **RK4 Learning and Timing Comparison**. In Section C.7 we show a comparison with the RK4 solver from torchdiffeq for learning longer noisy sine waves with 300 and 1000 steps and compare the times.
> The results (Table 3. Figures 14,15) show that the QP solver is able to learn significantly better fits and is up to 200x times faster in this experiment.
>
> However, if there are suggestions for any additional ablation experiments, please let us know.

---

> > ### Comment · Reviewer_5iBu · 2023-11-21
> >
> > Thank you for your responses which mitigate my concerns. I acknowledge the paper's novelty and appreciate the value of your insights to the community. To meet the rigorous standards of ICLR, I however agree that a thorough review of notations, descriptions, and theoretical guarantees is essential. Acknowledging limitations is a key aspect of scholarly work, and I suggest the authors clearly articulate these in their revision. Further refinement is indeed necessary. I recommend submitting an enhanced version to upcoming conferences or journals for consideration.
> >
> > I keep my current score.

---

### Official Review · Reviewer_Uvxo · 2023-10-31

**Soundness:** 3 good
**Presentation:** 3 good
**Contribution:** 3 good
**Rating:** 8
**Confidence:** 4

**Summary:**

In this paper the authors introduce a methodology wherein they learn a neural network to output the coefficients of an ODE instead of the solution in itself, hence making the output of the neural network more interpretable.

The key idea is that for the ODE written in the form of equation 2 in the paper (which is a general form that should comprise of both linear, nonlinear and time-varying PDEs), the authors propose to learn a neural network that outputs the coefficients of the ODE. The authors also point to the fact that their method can also be used to infer the (temporal) discretization of the ODE from observation data as well.

The learning of the neural networks approximating the ODE coefficients is done by solving a system of equations subject to quadratic constraints. The number of parameter that the network needs to approximate are determined by the number of time-grids and the order of the ODE.

**Strengths:**

The work provides a very interesting methodology a single governing equation of a linear/nonlinear system in an interpretable manner.

The authors show through their experiments that they are able to learn nonlinear ODEs (unlike previous work like SINDy).

The methodology also enables the authors to infer quantities like the temporal discretization and also the initial conditions gives a set of trajectories which is quite useful/interesting.

**Weaknesses:**

While the methodolgy is pretty interesting, I wonder how it scales with the number of dimensions in the input data, and for more complex systems like Navier-Stokes. Some discussion related to it would be useful!

The results on PDEs like Darcy Flow show that the network is not as much better (at least in performance) as compared to FNO baseline.

**Questions:**

- I am a bit unclear about how the authors are able to parallelize in terms of the sequences, I understand the parallel batchwise training aspect of the training. Is is that since we have ground truth training data, and the we can write down the forward and backward Taylor approximations, we get different sets of equations that are solved in parallel?
- It seems that the authors are solving a relatively complex system of equations stemming from the discretization of the ODEs. Are there any set of equations that may not be solvable due to the given methodology.

---

> ### Author Response · Authors · 2023-11-14
>
> We thank the reviewer for reviewing and appreciating our work. We address some of the points and questions raised below.
>
>
> >>I wonder how it scales with the number of dimensions in the input data, and for more complex systems like Navier-Stokes.
> >>
> >>The results on PDEs like Darcy Flow show that the network is not as much better (at least in performance) as compared to FNO baseline.
>
> **PDE**. In this paper we present a *general* framework for learning the governing equation from data and not a *specific* solution to a specialized class of PDEs. Naturally, when narrowing down the focus to PDE forecasting, a specialized solver like FNO works very well.
>
> MNN are a general and encompassing framework compared with the overarching state-of-the-art of ODE-related ML methods where
>
> 1. Neural ODEs model sequences but cannot discover governing equations,
> 2. SINDy discovers governing equations but is not with neural networks so it cannot scale and it cannot discover non-linearly entangled governing equations,
> 3. FNO can forecast future trajectories but is purely data driven, has no notion of governing equations.
>
> MNNs cut across these types of methods and in many cases perform better.
>
> **Scaling**. Scaling to large dimensions especially for PDEs depends on how effectively sparse matrix methods can be used for solving linear equations, since QP constraint matrices are highly sparse.
> Currently we use dense matrix factorizations which are fast and have manageable memory usage for the problems we have considered.
> We plan to investigate sparse matrix methods for PDEs in follow-up work.
>
>  >> I am a bit unclear about how the authors are able to parallelize in terms of the sequences, I understand the parallel batchwise training aspect of the training. Is is that since we have ground truth training data, and the we can write down the forward and backward Taylor approximations, we get different sets of equations that are solved in parallel?
>
> **Parallelization**. Yes, since we can write the discretized ODE over an entire time sequence in terms of the constraints, we only have to solve the constrained optimization to get the entire solution over the grid without stepping through time.
>
> One minor point is that the parallel solution does not depend on the ground-truth data and is a feature of the forward pass.
>
> >>It seems that the authors are solving a relatively complex system of equations stemming from the discretization of the ODEs. Are there any set of equations that may not be solvable due to the given methodology.
>
> Solving the equality-constrained quadratic program only requires solving linear equations so as long as the constraints are not inconsistent (such as contradictory initial conditions) we do not encounter any problems.
>
> However, as we mention in Section 3.1 regarding non-linear ODEs, the QP method on its own does not solve non-linear ODEs (since the constraints are linear).
> For this we introduce new variables representing non-linear terms (say a squared term) in the QP and solve the QP.
> The the solution corresponding to the non-linear variables is related to the original in the MSE loss function using the required non-linear function (say the square function).
>
>
> We thank the reviewer again for the review and appreciation and hope that we were able to answer the reviewer's questions.

---

### Author Response · Authors · 2023-11-14
**General Summary**

We thank the reviewers for their effort. We note that all reviewers appreciate the magnitude of the importance of what we try to achieve.

*Misunderstood limitations*.
We believe that our work has been quite misunderstood, partly due to the intersectional nature of the work.
The technical points raised by reviewers 5iBu and L7km are simple misunderstandings, and we detail them below.
We simplify notation and clarify where suggested.
We also include an error analysis of the solver in the new draft.
We thank the reviewers for raising the points, because in the end these would be the points of any reader in the community going through our paper, and urge the reviewers to revisit our paper.

*Significance of method*. We believe this a significant step in the Scientific Machine Learning literature, since to date and to best of our knowledge there is no general class of neural networks that inherently structures itself around governing equations instead of numerical tensor-based representations.

1. Neural ODEs model sequences but cannot discover governing equations and have significant performance limitations,
2. SINDy discovers governing equations but is not with neural networks so it cannot scale and it cannot discover non-linearly entangled governing equations,
3. FNO and related methods forecast future trajectories in purely supervised and data driven manner with no notion of governing equations.

By contrast, the proposed Mechanistic Neural Networks build in governing equations as symbols in the representational structure of the network and define learning mechanisms to discover the best fitting equations given data in an efficient, scalable, and interpretable manner.

The general approach is as follows.

1. A neural network taking input x produces an explicit ODE U(x) which is discretized over a grid.
2. The discretized ODE is solved by a differentiable fast, batch- and step-parallel solver with learnable steps.
3. The solution is used in a decoder.

We focus on scientific data, we run the (mechanistic) neural network on these scientific data and we can derive *equations* as outputs of the neural network (either in the hidden layers or the output at will).
The equations can be used either for interpretation and analysis (as with an equation derived by a scientist or SINDy) or for prediction (as with FNO, Neural ODEs etc).

Update: Example code for our work is available at: https://anonymous.4open.science/r/mech-nn-FFF4

---

### Meta-Review · Area_Chair_DtwN · 2023-12-12

**Metareview:**

The paper proposes an approach to learn more "intepretable" ODEs from data by fitting the coefficients of the ODE by neural networks rather than overall "solution map" that the ODE is describing. The idea is to express the ODE equality constraints and boundary conditions as a quadratic program, and use this parametrization in turn to use "implicit derivation" techniques ala Amos and Kolter. The performance is verified on some simple dynamics like Lorenz.

On the pro side, the idea in the paper is fairly clean, and at high level well-described. However, I agree with Reviewer 5iBu and L7km that the notation and mathematical exposition is confusing and hard-to-track at times --- with various quantities being overloaded or labeled differently in different parts of the paper. I also agree with Reviewer Uvxo that the experiments are for relatively simple systems, and it's unclear how scalable the strategy is even for moderately complex systems. Finally, I agree with Reviewer 5iBu that not enough is discussed in the paper about "sample complexity" limitations --- that is, the number of equations needed to effectively pin down the coefficients --- or at least, to point out cases in which this strategy is expected to fail.

**Justification For Why Not Higher Score:**

The notation and exposition in the paper is at times hard to track. The experiments are performed for relatively simple dynamical systems, and it's not clear how the approach scales even for moderately challenging systems. Finally, not enough is said about "sample complexity", or effective number of equations needed to pin down the coefficients and/or when the approach is expected to struggle.

**Justification For Why Not Lower Score:**

N/A

---

### Decision · Program_Chairs · 2024-01-16

Reject